# Transformers at a Fraction

Aritra Mukhopadhyay[1,2], Rucha Bhalchandra Joshi[1,2], Nidhi Tiwari[3], and Subhankar Mishra[*1,2]

[1]School of Computer Sciences, National Institute of Science Education and Research, Bhubaneswar, 752050, India
[2]Homi Bhabha National Institute, Mumbai, 400094, India
[3]Microsoft Ltd., India
aritra.mukhopadhyay@niser.ac.in, rucha.joshi@niser.ac.in, nidhitiwari@microsoft.com, smishra@niser.ac.in

## Abstract

Transformer-based large models, such as GPT, are known for their performance and ability to effectively address tasks. Transformer-based models often have many parameters, which are trained to achieve high-performance levels. As a result, they cannot be run locally on devices with smaller memory sizes, such as mobile phones, necessitating the use of these models remotely by sending the data to the cloud. This exposes us to privacy concerns over sending confidential data to the server, among others. In this work, we propose a method to make these large models easier to run on devices with much smaller memory while sacrificing little to no performance. We investigate quaternion neural networks, which can reduce the number of parameters to one-fourth of the original real-valued model when employed efficiently. Additionally, we explore sparse networks created by pruning weights as a method of parameter reduction, following the Lottery Ticket Hypothesis.

We perform the experiments on vision and language tasks on their respective datasets. We observe that pruned quaternion models perform better than the real-valued models in severely sparse conditions.

## 1 Introduction

The advent of powerful transformer-based models such as GPT-4 [1], LLaMA [2], Mistral [3][4], Vision Transformer (ViT) [5], ViTDet [6], and DINO [7] has revolutionized various domains, including natural language processing, computer vision, and image classification. However, the sheer scale of these models poses significant deployment challenges, with their massive parameter counts and computational requirements rendering them impractical for edge devices. For instance, running the smallest LLaMA 3 model (8B parameters) on a Raspberry Pi 5 (8GB) yields a meager 1-2 tokens per second (experiment done using ollama).

Hence, we take up addressing this challenge of reducing the number of parameters, thereby reducing the burden of tuning a large number of parameters.

Techniques like Mixture of Experts (MoE) models like Mixtral [4] and knowledge distillation have been explored to alleviate these challenges. MoEs selectively utilize multiple sub-models based on input data, enabling faster inference while keeping the output quality and total number of parameters high.

The pursuit of efficient model deployment has led to the exploration of various techniques, including quantization, which reduces model weight precision [8]. While effective, these methods often compromise on accuracy. On the other hand, pruning methods like SparseGPT [9] have demonstrated one-shot pruning to around 50-60% of the weights while maintaining similar performance. However, given the large size of transformer-based models, even a 50-60% reduction may not be sufficient to fully address the computational challenges they present.

To further reduce the number of parameters, we explore the application of the Lottery Ticket Hypothesis (LTH) [10] to transformers [11–13], and the use of quaternion-valued weights in transformers [14–16], both of which have been implemented separately in previous studies. For instance, LTH has been successfully applied to pre-trained BERT networks [12] and vision transformers [13], while quaternion networks have shown promise in various tasks such as facial expression recognition [16], NLP [14, 17], and hyperspectral image classification [15]. In this work, we combine these approaches, applying LTH to quaternion transformers, and make the following research contributions:

1. Apply Lottery Ticket Hypothesis (LTH) to Quaternion transformers.

2. Demonstrate that quaternion-valued lottery tickets perform better than real-valued lottery tickets at higher sparsities across various computer vision as well as natural language processing tasks.

3. Release a PyPi package called `qytorch`[1] for easy implementation of quaternion layers in PyTorch.

---

[*]Corresponding Author.

[1]https://pypi.org/project/qytorch/

Proceedings of the 6th Northern Lights Deep Learning Conference (NLDL), PMLR 265, 2025.

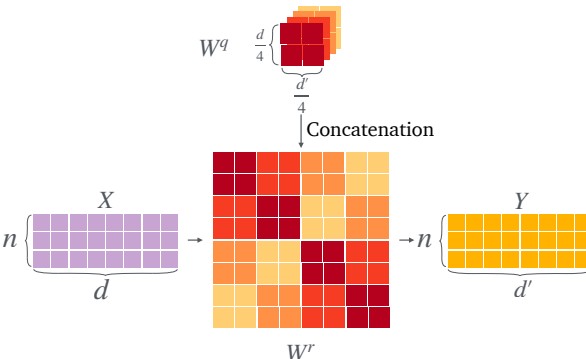

**Figure 1.** Construction of the real weight matrix from quaternion components

## 2 Background

In this section, we introduce quaternions and the Lottery Ticket Hypothesis, which serve as the foundation for our work.

### 2.1 Transformer Self Attention

$$f(x) = \text{softmax}\left(\frac{(x \cdot W_q^r) \cdot (x \cdot W_k^r)^T}{\sqrt{d'}}\right) \cdot (x \cdot W_v^r) \tag{1}$$

The self-attention mechanism in transformers computes a weighted representation of the input by first generating the Query ($Q$), Key ($K$), and Value ($V$) matrices through linear transformations of the input $x \in \mathbb{R}^{(n,d)}$ using three real weight matrices, $W_Q^r$, $W_K^r$, and $W_V^r \in \mathbb{R}^{(d,d')}$. Attention scores, calculated as the cosine similarity between each vector in $Q$ and each vector in $K$, are obtained by multiplying the $Q$ and $K$ matrices, followed by normalization and the application of the softmax function. The resulting values are then multiplied by the $V$ matrix to produce the final output. This process is summarized by the equation 1.

### 2.2 Quaternion Networks

Quaternions are mathematical constructs widely used in computer graphics, simulations, and other applications. These can be represented as four-dimensional extensions of complex numbers, expressed as $q = a + bi + cj + dk$, where $a, b, c$, and $d$ are real numbers, and $i, j, k$ are imaginary components. In neural networks, quaternion representations reduce computational requirements. Calculations are facilitated by representing quaternions in matrix notation, as shown in Eq 2.

$$q = \begin{bmatrix} a & -b & -c & -d \\ b & a & -d & c \\ c & d & a & -b \\ d & -c & b & a \end{bmatrix} \tag{2}$$

As illustrated in Figure 1, using quaternions to represent the weights reduces the number of parameters by four times while maintaining performance at least as good or sometimes even better [18]. A real weight matrix $W^r$ with $m \times n$ weights, where $m$ and $n$ are multiples of 4, can represent four $\frac{m}{4} \times \frac{n}{4}$ dimensional quaternion weight matrices $W^{q_r}, W^{q_i}, W^{q_j}$ and $W^{q_k}$ (representing $r$, $i$, $j$, and $k$ components of quaternion weight matrix $W^q$). This results in a total of $4 \times \frac{m}{4} \times \frac{n}{4} = \frac{m \times n}{4}$ weights, which is four times fewer than the real-valued representation. $W^{q_r}, W^{q_i}, W^{q_j}$ and $W^{q_k}$ matrices are concatenated as shown in Eq 3 and Fig 1 to create the $W^r$.

$$W^r = \begin{bmatrix} W^{q_r} & W^{q_i} & W^{q_j} & W^{q_k} \\ W^{q_i} & W^{q_r} & W^{q_k} & W^{q_j} \\ W^{q_j} & W^{q_k} & W^{q_r} & W^{q_i} \\ W^{q_k} & W^{q_j} & W^{q_i} & W^{q_r} \end{bmatrix} \tag{3}$$

### 2.3 Lottery Ticket Hypothesis

The Lottery Ticket Hypothesis (LTH) proposed by Frankle et al. [10] states that within a randomly initialized neural network, there exists a subnetwork that can match the original network's test accuracy when trained in isolation. This subnetwork, called the "winning ticket", achieves comparable accuracy after training for at most the same number of iterations as the original network. Mathematically, this means that for a network $Q$ with minimum validation loss $l_1$ at iteration $j_1$, there exists a subnetwork $Q'$ with fewer parameters that reaches its minimum validation loss $l_2$ at iteration $j_2$, where $j_2 \leq j_1$ and $l_2 \leq l_1$.

## 3 Methodology

We designed custom Quaternion Layers for Linear, Convolutional, Self Attention, and Transformer Layers, mirroring the functionality of their PyTorch real-valued counterparts. This is released as a PyPi package called `qytorch`. For our experiments, we developed quaternion versions of Vision Transformer (ViT) and nanoGPT using `qytorch`, which allowed us to maintain consistency with the real counterpart of the existing architecture and implementation.

In the context of Quaternion Transformers, the self-attention mechanism involves three weights: Query ($W_Q^q$), Key ($W_K^q$), and Value ($W_V^q$). Each of the three contain 4 small weight matrices corresponding to the $r$, $i$, $j$ annd $k$ components of a quaternion as shown in Fig 2. Using these 12 quaternion weights ($W_{Q,K,V}^q$) we construct the 3 real weight matrices ($W_{Q,K,V}^r$) following Fig 1.

We conducted a series of reset-train-prune experiments using L1 unstructured pruning. At each pruning iteration, a fraction $p$ (prune rate) of the weights in the $r, i, j, k$ components of the three $W_{Q,K,V}^q$

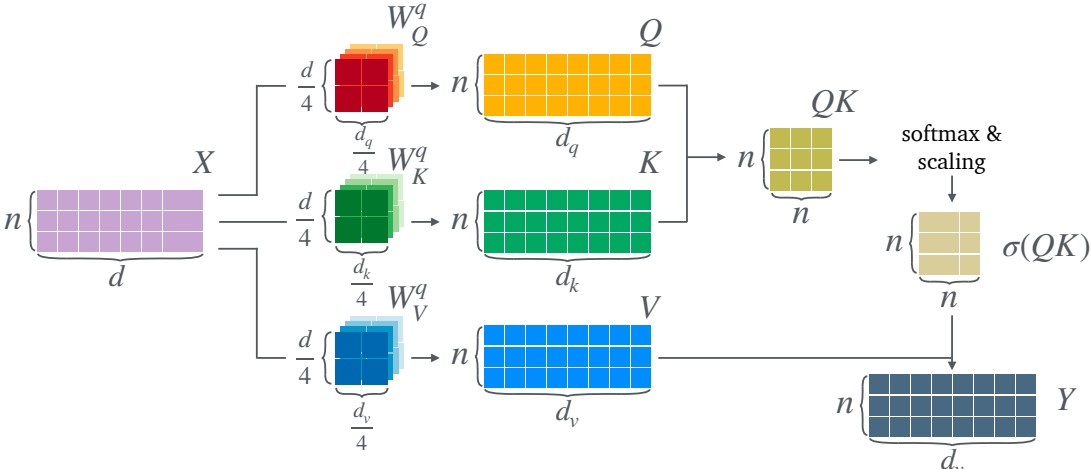

**Figure 2.** Self attention in quaternion setting

weights was pruned. Each of the 12 matrices were pruned independently. This process, detailed in Algorithm 1, was repeated $n_p$ times. By fixing the seed before reinitializing the model, we ensure that the model parameters are reset to the same values each time. The pruning was designed such that $(1-p)^{n_p}$ resulted in a sparsity level below 1%.

---
**Algorithm 1** Reset-Train-Prune Process for Transformers

---
Set the value $p$ ($p \in (0,1)$)
**for** $i$ in 0 to $n_p - 1$ **do**
    1. Fix seed and **Reinitialize** model parameters
    2. **Train** model for required number of epochs
    3. **Prune the** $W_Q^q, W_K^q, W_V^q$ **weight matrices**, removing a **fraction p** of the weights compared to what was left before pruning
**end for**

---

## 3.1 Implementation Details

- **Quaternion:** Following [19, 20], we kept the norm layers, bias, and embedding and class tokens (in ViT) as real-valued, since they collectively contribute less than 1% to the total number of parameters. Moreover, norm weights and bias being 1D arrays, we generally cannot convert them to quaternion matrices which require last two dimentions to be multiple of four. Additionally, existing literature on quaternion-based norm layers are slow and do not reduce parameter counts.

- **Pruning:** We strategically excluded specific layers from pruning, including norm layers, biases, positional embedding, and the class token in Vision Transformer (ViT), as they barely contribute to the overall model size. But in case of

| Task | Model | Dataset | #params Real | #params Quat | Prune Rate |
|------|-------|---------|------|------|-----------|
| Vision | ViT | CIFAR10 | 1.1M | 0.3M | 0.4 |
| | | CIFAR100 | 1.2M | 0.3M | 0.4 |
| | | MedMNIST | 28.95M | 7.72M | 0.3 |
| Language | nanoGPT | Shakespeare | 10.65M | 2.79M | 0.3, 0.5, 0.7 |

**Table 1.** Summary of all the experiments performed, the models used, their parameter counts, and the pruning percentages used for each.

smaller models like our ViT (4.1.1), these layers account for approximately 1-2% of the model's parameters, which although becomes significant in extremely sparse conditions, do not cause much problems. We experimented with pruning norm weights and bias weights, observing that pruning norm weights hurts performance more than pruning bias weights.

- **Resetting:** All the weights and biases were reset without any deviation.

## 4 Experiments

In this section, we give the details of the experiments we conducted. We explored the two most popular tasks where transformers frequently find use in Computer Vision and Language Modelling. On the vision front, we tried the well-known model ViT, which generally has many parameters and performs pretty well in various vision tasks. To evaluate the performance of our approach on the language task, we use the nanoGPT model. The summary of the experiments performed, the models used, and their parameter count has been provided in table 1.

## 4.1 Computer Vision

We performed the experiments to show the efficiency of our proposed method on vision tasks. The pri-

| Dataset | Classes (with dummies) | Number of Channels | Number of Train Images |
|---|---|---|---|
| PathMNIST | $9 \rightarrow 12$ | 3 | 89996 |
| DermaMNIST | $7 \rightarrow 8$ | 3 | 7007 |
| OCTMNIST | $4 \rightarrow 4$ | 1 | 97477 |
| PneumoniaMNIST | $2 \rightarrow 4$ | 1 | 4708 |
| RetinaMNIST | $5 \rightarrow 8$ | 3 | 5368 |
| BreastMNIST | $2 \rightarrow 4$ | 1 | 546 |
| BloodMNIST | $8 \rightarrow 8$ | 3 | 11959 |
| TissueMNIST | $8 \rightarrow 8$ | 1 | 165466 |
| OrganAMNIST | $11 \rightarrow 12$ | 1 | 34561 |
| OrganCMNIST | $11 \rightarrow 12$ | 1 | 12975 |
| OrganSMNIST | $11 \rightarrow 12$ | 1 | 13932 |

**Table 2.** Summary of the used MedMNIST datasets.

mary task we address is the classification on image datasets, CIFAR10, CIFAR100 and different datasets in MedMNIST. The summary of these datasets is given in the table 2.

### 4.1.1 Vision Transformer on CIFAR10 and CIFAR100

We modified the Vision Transformer (ViT) [21] models for our experiments, adjusting the number of transformer layers and embedding dimensions to optimize accuracy. For the small CIFAR10 dataset, we used a compact model with 6 transformer layers, an embedding dimension of 64, 8 heads, and an MLP size of 512, resulting in approximately 1.2 million initial parameters in real for both CIFAR10 and CIFAR100. The quaternion-based model had around 0.3 million parameters. We observed that, ViT models easily overfit on small datasets like CIFAR, which is why we took this approach. Figures 3(a) and 3(b) illustrate the performance of our model on the CIFAR10 and CIFAR100 datasets.

Due to the quaternion layer limitation that input and output neurons must be a multiple of 4 [22–24], we added two dummy classes to the CIFAR10 dataset, making the model output three quaternions corresponding to 12 classes. This approach had no distinguishable impact on the learning statistics but allowed us to use purely quaternion models.

### 4.1.2 Vision Transformer on MedMNIST

We conduct experiments on the MedMNIST dataset [25] [26], a comprehensive repository of 12 medical image datasets. The dataset's diversity, featuring varying numbers of classes, images, and channels, makes it well-suited for evaluating our models' performance under different conditions.

We use the Vision Transformer (ViT) model with modified specifications: 8 transformer blocks, each with 8 attention heads, a ViT patch size of 7, self-attention embedding dimension of 768, prediction head MLP size of 768, and dropout regularization with a rate of 0.1.

We deliberately designed the model to have a lower weight count for the MLP layers compared

to the transformer weights. Initially, we found that the MLP had a significant weight count in the ViT-Base configuration [5]. We reduced the MLP size to address this, minimizing its contribution to the total weight count. While increasing transformer properties (e.g., number of heads or layers) didn't significantly impact accuracy, boosting MLP layer characteristics substantially improved performance. However, we prioritized transformer characteristics, maintaining a low MLP size and high transformer properties.

The results, as shown in Figures 3(c) and 3(d), are consistent with our MedMNIST experiments, demonstrating that the quaternion model's performance remains high till a much higher sparsity level compared to the real-valued model.

### 4.2 Language Modeling

We implement Quaternion and LTH pruning on the NanoGPT model using the Shakespeare [27] dataset, which contains 40,000 lines of Shakespeare's plays. The nanoGPT model [28] generates the next most probable token based on a given context.

We trained the nanoGPT model with original hyperparameters and seeds to reproduce the results from the original work. The dataset has approximately 64 characters (rounded to a multiple of 4), where each character is treated as a token.

The full real model has about 10.65M parameters, while the quaternion version has about 2.7M parameters. We prune the nanoGPT model using three rates: 0.3, 0.5, and 0.7.

### 4.2.1 Perplexity Score

Perplexity score is calculated as $PPL(X) = \exp(-\sum p(x) \log q(x))$ ($p(x)$ and $q(x)$ denotes output and true probabilities respectively). A lower perplexity score indicates better performance, while a higher score suggests that the model is less accurate or more "perplexed" by the query.

## 5 Results and Discussion

### 5.1 Computer Vision

Figures 3 show that the quaternion model's performance is comparable to the real-valued model inspite of having one-fourth the number of weights, achieving similar accuracy with fewer parameters. Additional results confirming similar observations are also shown in figure 9 in Appendix which contains our results related to the MedMNIST dataset.

In figure 4, we observe the different values of weights and their frequency of occurrence during pruning iterations. The insignificant weights closer to zero are pruned, revealing the effective weights

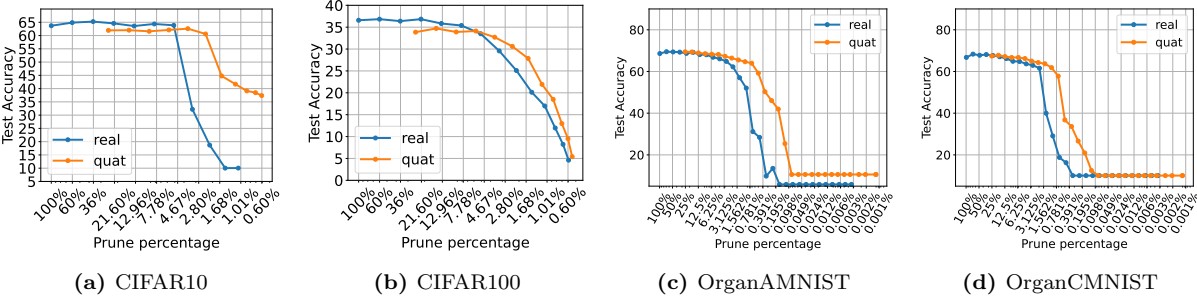

**(a)** CIFAR10     **(b)** CIFAR100     **(c)** OrganAMNIST     **(d)** OrganCMNIST

**Figure 3.** Graph of highest accuracy achieved on full training after each pruning in real and quaternion versions of ViT on CIFAR10, CIFAR100 and two medMNISTs. Other results are in appendix.

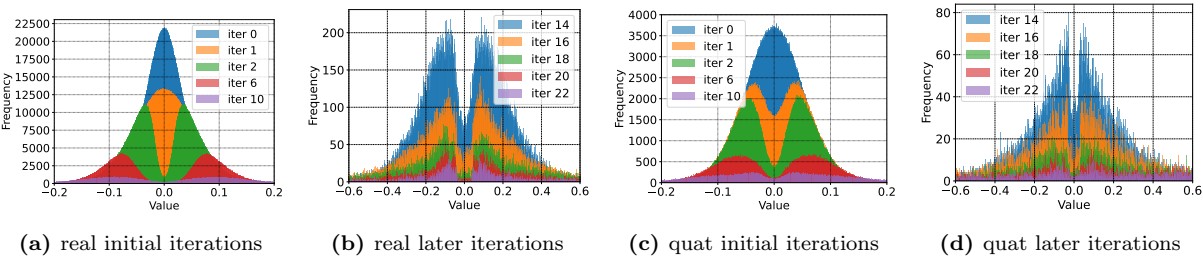

**(a)** real initial iterations    **(b)** real later iterations    **(c)** quat initial iterations    **(d)** quat later iterations

**Figure 4.** trained nanoGPT weight distribution across pruning iterations for real and quat on Shakespeare dataset

contributing to inference. As we prune the model, the number of remaining weights decreases significantly. The near-zero weights are pruned in every iteration, making the distribution two peaks away from zero. Similar results can be seen in figure 6, which corresponds to the MedMNIST dataset.

## 5.2 Language Modeling

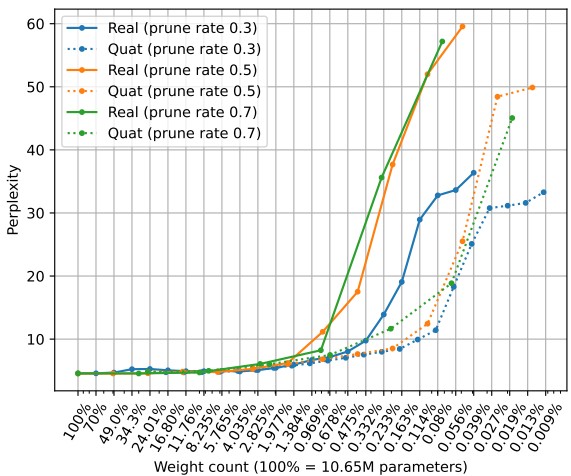

**Figure 5.** Perplexity scores of real and quaternion models at different pruning iterations, shown with respect to the percentage of weights remaining relative to the original number of weights in the unpruned real-valued model.

The performance of the nanoGPT model on the Shakespeare dataset for all prune rates is shown in

Figure 5 using perplexity scores. The figure compares the perplexity scores of quaternion and real-valued models with respect to the percentage of weights remaining relative to the unpruned real-valued model. It demonstrates that quaternion-valued models, which originally have approximately four times fewer weights, maintain their performance even when further pruned, resulting in lower perplexity scores compared to the real-valued versions.

## 5.3 ViT on MedMNIST weight distributions graphs

Figure 6 displays the trained weight distributions of Vision Transformer (ViT) models on two MedMNIST datasets, PathMNIST and OCTMNIST, across different pruning iterations. The histograms show the number of weights whose values fall within each bin.

In unpruned models, a significant portion of weights are concentrated near zero, indicating they don't contribute significantly to predictions. As we prune, the remaining weights adjust themselves, converging to a value not very close to zero. This suggests the model finds an alternative optimum that doesn't rely on very small weights.

Quaternion weights exhibit a higher standard deviation than real-valued counterparts, indicating diverse values. They respond better to pruning, with larger middle gap sizes in quaternion models compared to real models.

Unlike Figure 4, we don't see a clear Gaussian distribution. Individual ViT layers follow Gaussian distributions with mean close to zero but different

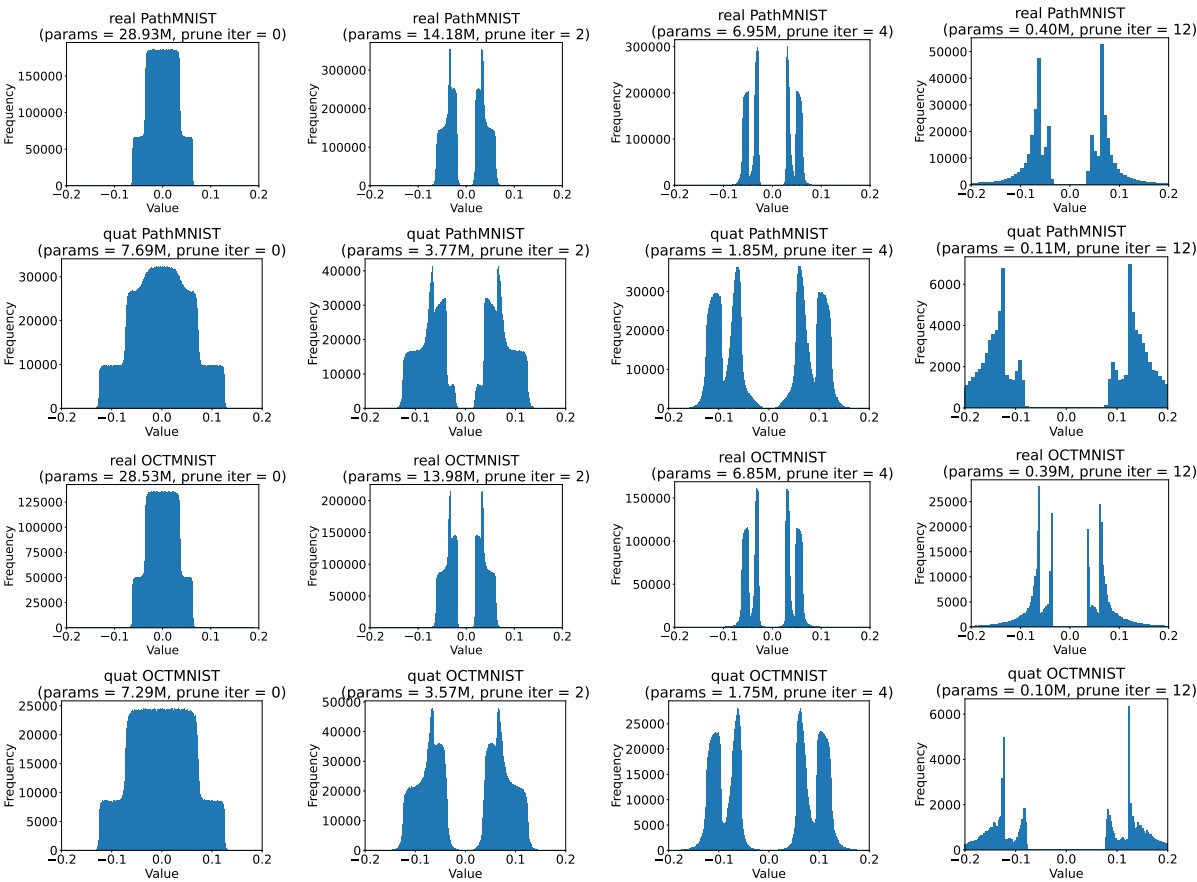

**Figure 6.** Trained ViT weight distribution across pruning iterations for real and quaternion models on two MedMNIST datasets. The columns show the weight distribution for unpruned, prune iteration 2, prune iteration 4, and prune iteration 12 in order. The rows stand for PathMNIST real, PathMNIST quaternion, OCTMNIST real, and OCTMNIST quaternion models respectively.

standard deviations. The sum of these distributions results in a non-Gaussian overall weight distribution, more pronounced in the third column (prune iteration 4), where two distinct peaks are visible.

This trend is consistent across all MedMNIST datasets and CIFAR datasets used in our experiments.

# 6 Conclusion

In this work, we explore the use of Quaternion variations in Transformer models to reduce weight parameters by one-fourth compared to traditional real-valued Transformers. We also test the effectiveness of the Lottery Ticket Hypothesis in pruning these Quaternion-based models and compare their performance with real-valued Transformers.

One limitation of our approach is that Quaternion neural networks require the number of input and output features to be multiples of four. While this constraint is generally not problematic for hidden layers, as large neural networks often use feature sizes that are large powers of two, it can pose a challenge for input and output layers where feature sizes

may not align with this requirement. We are actively exploring methods to overcome this limitation, ensuring broader applicability of Quaternion-based models across different layers and architectures.

Looking ahead, combining our approach with methods like quantization [8], pruning [29–31], and a mixture of experts [32] can lead to even more efficient and deployable models that maintain performance on various devices. We are also interested in investigating the impact of other complex space transformations of weights, such as octonions, and the direct conversion of real trained weights to quaternions without or with minimal training.

# Acknowledgements

This work was partially funded by the Microsoft Academic Partnership Grant (MAPG) 2023 and NISER-RIN4001.

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

# 7 Appendix

In this section, we expand on the experiments conducted in the paper. Further results on nanoGPT and vision transformer on MedMNIST are reported here.

## 7.1 nanoGPT perplexity by iteration

The performance of the nanoGPT model during each pruning iteration is detailed in Figure 7. This figure compares the perplexity scores of quaternion- and real-valued models at each iteration, illustrating how the perplexity score increases as more weights are pruned. The results highlight that, despite an increase in perplexity due to the pruning, quaternion-valued models tend to maintain lower perplexity scores compared to real-valued models, particularly in later iterations. This behavior suggests that quaternion models are more resilient to pruning, maintaining performance even as the sparsity level increases.

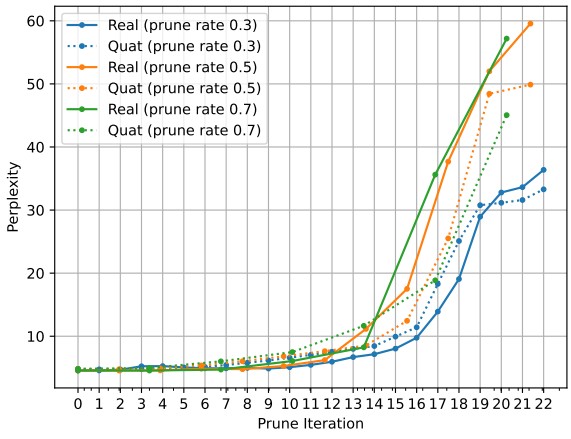

**Figure 7.** Perplexity scores (y-axis) of real and quaternion models across different pruning iterations (x-axis) for the nanoGPT model.

## 7.2 MedMNIST real vs quat performance

The results of the MedMNIST experiments are compiled in Figure 9, which displays the highest accuracy achieved on the full training dataset after each pruning iteration. The figure compares the performance of the real-valued and quaternion versions of the Vision Transformer (ViT) across various MedMNIST datasets, using a prune rate of 0.3. The consistent patterns observed across all experiments further reinforce the resilience of quaternion models, as they tend to maintain higher accuracy levels compared to their real-valued counterparts, even as pruning progresses.

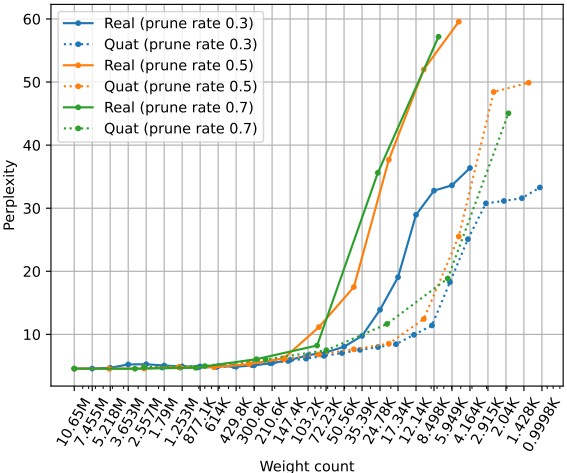

**Figure 8.** Perplexity scores of real and quaternion models at different pruning iterations, shown with respect to the exact number of weights remaining.

## 7.3 Demo Code to implement Quaternion Self Attention Layer in pytorch

Here I have implemented the Quaternion Self Attention Layer (`QSelfAttension`) using Pytorch. Here we have two helper functions: `self_attention` and `quat_to_real`. The `self_attention` function is used to calculate the self-attention given the input tensor and the real weights. The `quat_to_real` function is used to convert the quaternion weights to real weights. Finally there is the `QSelfAttention` class which is used to implement the quaternion self attention layer.

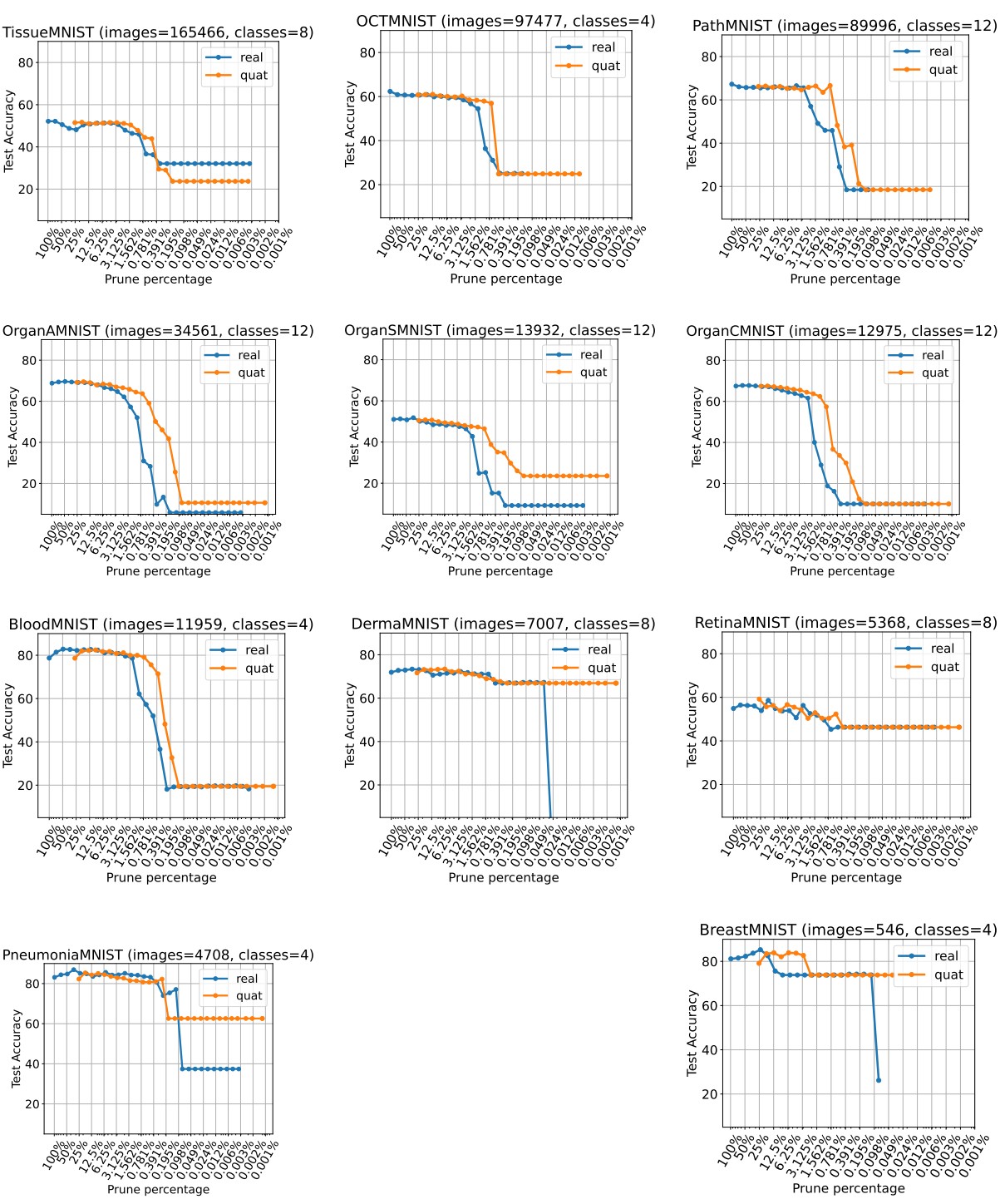

**Figure 9.** Graph of Highest Accuracy achieved on full training, after each pruning, in real and quat versions of Vision Transformer (ViT) trained on the MedMNIST dataset with a prune rate of 0.3.

```python
import torch
import torch.nn.functional as F
import torch.nn as nn
import math

def self_attention(x, Wq=None, Wk=None, Wv=None):

    Q = x @ Wq if Wq is not None else x
    K = x @ Wk if Wk is not None else x
    V = x @ Wv if Wv is not None else x

    d_prime = Wq.shape[1] if Wq is not None else x.shape[1]

    attention_weights = F.softmax(Q @ K.T / d_prime**0.5, dim=-1)
    output = attention_weights @ V

    return output

def quat_to_real(r, i, j, k):
    return torch.cat([torch.cat([r, -i, -j, -k], dim=0),
                      torch.cat([i,  r, -k,  j], dim=0),
                      torch.cat([j,  k,  r, -i], dim=0),
                      torch.cat([k, -j,  i,  r], dim=0)], dim=1)

class QSelfAttension(nn.Module):
    def __init__(self, in_features: int, out_features: int) -> None:
        assert in_features % 4 == 0 and out_features % 4 == 0, "in_channels and
    out_channels must be divisible by 4"

        super().__init__()
        self.in_features = in_features
        self.out_features = out_features

        for QKV in ["Q", "K", "V"]:
            for rijk in ["r", "i", "j", "k"]:
                self.register_parameter(
                    f"{QKV}{rijk}",
                    nn.Parameter(torch.empty((out_features//4, in_features//4)))
                )
        self.reset_parameters()

    def reset_parameters(self) -> None:
        for QKV in ["Q", "K", "V"]:
            for rijk in ["r", "i", "j", "k"]:
                nn.init.kaiming_uniform_(
                    getattr(self, f"{QKV}{rijk}"),
                    a=math.sqrt(5)
                )

    def forward(self, x: torch.Tensor) -> torch.Tensor:
        Wq_real = quat_to_real(self.Qr, self.Qi, self.Qj, self.Qk)
        Wk_real = quat_to_real(self.Kr, self.Ki, self.Kj, self.Kk)
        Wv_real = quat_to_real(self.Vr, self.Vi, self.Vj, self.Vk)
        return self_attention(x, Wq_real, Wk_real, Wv_real)
```

