# OpenReview forum: "Transformers at a Fraction"
_NLDL.org/2025/Conference — NLDL 2025 Oral_

### Official Review · Reviewer_xFiY · 2024-10-01
**A good paper which needs more details about the theory behind the method**

**Confidence:** 5

**Summary:**

The authors present a new deep learning method using quaternions to improve computation and memory requirements for use on smaller, local hardware. They also show that their method can be combined with the Lottery Ticket Hypothesis to prune the neural network and achieve even smaller neural networks, requiring even less memory usage.

**Strengths:**

The paper reads well and compares their approach thoroughly with standard – real-numbered – neural networks on vision and text tasks. Deep Learning methods are challenging to bring into settings where GPU/accelerated hardware is not available (e.g., handheld, medical and remote devices) and the combinations of the two methods presented by the authors are promising.
I enjoyed that the pruning is tested on both real and quaternions models, which acts as an ablation study.

**Weaknesses:**

The authors focus on parameter count, but there is no description and testing of memory and computation requirements at runtime, under the form of FLOPS, and MB of (V)RAM being used.

There is a description of the quaternion model in part 2 as well as a figure, but a mathematical description of the method is lacking (i.e., an equation or algorithm showing the concatenation of the q matrices to obtain W^r multiplied with the input). If space is needed, Algorithm 1 could be removed, as those operations are fairly straightforward and aligned with the LTH paper.

Although the paper states that only 25% of the parameters are required when using their approach (thus reducing the model weight), the computation may still require the same amount of used memory and computations, as the W^r matrix needs to be computed. Perhaps some optimizations were made, where the structure of W^r is exploited to enable faster computations and a smaller memory footprint, but this is not mentioned in the text.

Minor comments:
L015-L016: The general case of quaternion neural networks do not necessarily achieve a weight reduction of 75%, the authors purposely reduce the number of used parameters by structuring the weight matrix accordingly.
L087: Cosine similarity is perhaps a misnomer here, as the values are in [0,1] instead of [-1,1]. Dot product is unnormalized (what is being done) rather than the cosine function.
L105: I think it is rather confusing to use r,i,j,k as coefficient names (L114) instead of a,b,c,d as mentioned in L099.
L109: there is a "." before the citation.
Figure 1: Add the row/column sizes of the W^r matrix; The W^r matrix seem to have only four quaternions being reused at every row in a rolling pattern, but more colors should be used to highlight the fact that each q block is indeed different.

**Justification:**

I have been involved in work requiring pruning neural networks before, as well as work related to quaternions, although not directly linked to deep learning as in this paper.

---

> ### Author Rebuttal · Authors · 2024-10-22
>
> We thank the reviewer for their valuable feedback.
>
> > The authors focus on parameter count, but there is no description and testing of memory and computation requirements at runtime, under the form of FLOPS, and MB of (V)RAM being used.
>
> We thank the reviewer for the observation.
> The current implementation of pruning in PyTorch operates as a mask, which is also the basis for our library, qytorch. As a result, we do not currently observe significant gains in memory and computational requirements. However, with the available PyTorch implementations, we anticipate that future work will allow us to benefit from improvements in memory and computation efficiency which is proportional to the number of parameters.
>
> [1] https://pytorch.org/tutorials/intermediate/pruning_tutorial.html
>
> > There is a description of the quaternion model in part 2 as well as a figure, but a mathematical description of the method is lacking (i.e., an equation or algorithm showing the concatenation of the q matrices to obtain W^r multiplied with the input). If space is needed, Algorithm 1 could be removed, as those operations are fairly straightforward and aligned with the LTH paper.
>
> Thank you for the comment. We have updated Section 2.2 and added equation 3 to describe the concatenation of the q matrices to obtain the W^r as advised.
>
> > Although the paper states that only 25% of the parameters are required when using their approach (thus reducing the model weight), the computation may still require the same amount of used memory and computations, as the W^r matrix needs to be computed. Perhaps some optimizations were made, where the structure of W^r is exploited to enable faster computations and a smaller memory footprint, but this is not mentioned in the text.
>
> We appreciate the reviewer's observation regarding the computation and memory requirements. While the quaternion model reduces the model size to 25% of the original parameters, the computational complexity and memory usage may remain similar due to the need to compute the W^r matrix. This characteristic is common in quaternion networks. It gives us the advantage of more stability in pruning.
>
> We recognize the potential for optimizations leveraging the structure of W^r to improve efficiency and plan to explore this in future work. Thank you for your valuable feedback.We have added it as future work in our paper.
>
> >L015-L016: The general case of quaternion neural networks do not necessarily achieve a weight reduction of 75%, the authors purposely reduce the number of used parameters by structuring the weight matrix accordingly.
>
> Thank you for your comment. We reduce the number of neurons by one-fourth in the quaternion NN compared to real NNs similar to implementations in [1,2]. As a result, the parameter count reduces by one-fourth.
>
> > L087: Cosine similarity is perhaps a misnomer here, as the values are in [0,1] instead of [-1,1]. Dot product is unnormalized (what is being done) rather than the cosine function.
>
> Thank you for the comment. We have removed the reference to cosine similarity as correctly advised.
>
> > L105: I think it is rather confusing to use r,i,j,k as coefficient names (L114) instead of a,b,c,d as mentioned in L099.
>
> Thank you for the comment. We have updated the L105 with a,b,c,d as correctly advised.
>
> > L109: there is a "." before the citation.
>
> Thank you for the comment. We have updated the L099 as correctly advised.
>
> > Figure 1: Add the row/column sizes of the W^r matrix; The W^r matrix seem to have only four quaternions being reused at every row in a rolling pattern, but more colors should be used to highlight the fact that each q block is indeed different.
>
> Thank you for the comment. We have added the dimensions as advised. You have correctly pointed at the reuse of the four quaternions as the current implementation of quaternion transformers follow this practice [2,3]
>
> Please refer to the revised manuscript to see the updates.
>
> **References:**
>
> [1] Zhu, Xuanyu, et al. "Quaternion convolutional neural networks." Proceedings of the European conference on computer vision (ECCV). 2018.
>
> [2] Parcollet, Titouan, Mirco Ravanelli, Mohamed Morchid, Georges Linarès, Chiheb Trabelsi, Renato De Mori, and Yoshua Bengio. "Quaternion Recurrent Neural Networks." In International Conference on Learning Representations (ICLR) , 2019.
>
> [3] Chase Gaudet, Anthony Maida (2018). Deep Quaternion Networks. 2018 International Joint Conference on Neural Networks (IJCNN).

---

### Official Review · Reviewer_Xg9j · 2024-10-07
**Review for transformers at a fraction**

**Confidence:** 3

**Summary:**

This manuscript proposes a method to reduce the number of parameters so that the transformer-based large models can be used in on-device environments without significant performance degradation. By constructing the weight matrices of query, key, and value of transformer from quaternion components, the authors decrease the parameters of the transformer by $1\over4$. Also, they further diminish the number of parameters by pruning based on the Lottery Ticket Hypothesis (LTH). They demonstrate that their model shows similar or slightly lower performance than the original models with much fewer parameters on computer vision and language modeling tasks using Vision Transformer (ViT) and nanoGPT.

**Strengths:**

By simply applying the quaternion method, the number of parameters of the transformer is reduced to $1 \over 4$ without any decrease in performance. \
Through the quaternion method, the performance degradation is less or similar to the original model when the prune percentage is high (Fig. 3, 7), and the diversity of the pruned weights is also increased (Fig 4, 8). \
They distribute the PyPi package of quaternion transformer to help with follow-up research.

**Weaknesses:**

The quaternion method you applied requires that the real weight matrix be divisible by a multiple of 4, so adjustments such as dummy class padding are required. \
If the number of parameters after pruning is provided, the parameter reduction performance can be shown more clearly. \
What criteria were used to select the prune rate $p$, such as $p=0.4$ for ViT on CIFAR10 datasets? \
The models used in the experiments are insufficient to be called large models. It was good to show the results for each task, but it is unclear whether their method can work sufficiently in a large model. \
The contribution is not enough. Methods for reducing the parameters of the transformer based on the quaternion method have been proposed previously [1], and also LTH-based pruning methods are previously used [2, 3]. \
Furthermore, the proposed method requires iterative pruning with $n_p$ steps since it is based on the initial version of LTH-based pruning, which increases the model learning time. If you use the methods for obtaining ‘winning tickets’ faster [4, 5], the efficiency of your model can be improved. \
It is difficult to fully understand what the authors claim, like model pruning or parameter reductions from the current title.

Minor
* Table 1: There is no result about OpenWebText dataset
* (typo) line 259: “CIFAR” -> “MedMNIST”
* (typo) line 227: Prune rate might be “0.3, 0.5, and 0.3” -> “0.3, 0.5, and 0.7”
* The reference page needs to be rearranged

[1] Tay, Y., Zhang, A., Tuan, L. A., Rao, J., Zhang, S., Wang, S., ... & Hui, S. C. (2019). Lightweight and efficient neural natural language processing with quaternion networks. arXiv preprint arXiv:1906.04393. \
[2] Morcos, A., Yu, H., Paganini, M., & Tian, Y. (2019). One ticket to win them all: generalizing lottery ticket initializations across datasets and optimizers. Advances in neural information processing systems, 32. \
[3] Yu, H., Edunov, S., Tian, Y., & Morcos, A. S. (2019). Playing the lottery with rewards and multiple languages: lottery tickets in rl and nlp. arXiv preprint arXiv:1906.02768. \
[4] Lee, N., Ajanthan, T., & Torr, P. H. (2018). Snip: Single-shot network pruning based on connection sensitivity. arXiv preprint arXiv:1810.02340. \
[5] Wang, C., Zhang, G., & Grosse, R. (2020). Picking winning tickets before training by preserving gradient flow. arXiv preprint arXiv:2002.07376.

**Final Rebuttal Confidence:**

3

**Final Rebuttal Justification:**

The authors explain the weaknesses I pointed out well, and their arguments are valid. The lacking experiments have been supplemented in the rebuttal, and their ideas are novel.

**Justification:**

Their method can stably reduce many parameters while maintaining performance by replacing the weight matrices of the transformer with quaternion. Their method shows similar performance with fewer parameters in small transformer-based models, but it has not been proven whether it works well in large models. Nevertheless, it can be widely used for the models that utilize the transformer with less parameters and has scalability by providing a PyPi package.

---

> ### Author Rebuttal · Authors · 2024-10-22
>
> We thank the reviewer for their valuable feedback.
>
> > The quaternion method you applied requires that the real weight matrix be divisible by a multiple of 4, so adjustments such as dummy class padding are required.
>
> Thank you for your comment. This is a known limitation of using quaternions in general. We have noted the same in the *Conclusion* section.
>
> > If the number of parameters after pruning is provided, the parameter reduction performance can be shown more clearly.
>
> Thank you for the comment. We had followed the convention generally used by papers on Lottery Ticket Hypothesis. However, as suggested we have now added a new graph *Figure 7* in the appendix that shows the number of parameters (reduction in parameters) corresponding to *Figure 5*. We shall add corresponding graphs in camera ready to other Figures as well.
>
> > What criteria were used to select the prune rate p, such as p=0.4 for ViT on CIFAR10 datasets?
>
> Thank you for your comment. We performed experiments with different prune rates for CIFAR10 and other datasets, which resulted in similar trends in performance. However, only one representative case of prune rate p=0.4 is presented in the paper due to space constraints. We have added the same explanation to the paper for readers.
>
> > The models used in the experiments are insufficient to be called large models. It was good to show the results for each task, but it is unclear whether their method can work sufficiently in a large model.
>
> Thank you. As you have correctly observed, we want to show that our proposal works for a variety of transformer-led tasks. However, because of the limited computational resources, we are unable to verify our proposal for the larger transformer-based models.
>
> > The contribution is not enough. Methods for reducing the parameters of the transformer based on the quaternion method have been proposed previously [1], and also LTH-based pruning methods are previously used [2, 3].
>
> Thank you for your observation. In our paper, we propose applying the Lottery Ticket Hypothesis (LTH) to Quaternion Transformers, which results in a significantly greater reduction in parameters compared to using Quaternion Transformers alone. Additionally, our method demonstrates more stability, achieving higher accuracy at higher levels of sparsity compared to LTH applied to Real Transformers.
>
> > Furthermore, the proposed method requires iterative pruning with np steps since it is based on the initial version of LTH-based pruning, which increases the model learning time. If you use the methods for obtaining ‘winning tickets’ faster [4, 5], the efficiency of your model can be improved.
>
> We thank the reviewer for the suggestion and acknowledge the increase in model learning time. We shall explore the suggested methods in future to reduce the model learning time.
>
> > It is difficult to fully understand what the authors claim, like model pruning or parameter reductions from the current title.
>
> Thank you for your feedback. Our primary goal is achieving significant pruning which results in parameter reductions while maintaining strong performance. This is made possible by applying Lottery Ticket Hypothesis to Quaternion Transformers, which allows for more aggressive pruning compared to Real Transformers, without sacrificing accuracy. If we are allowed to update the title, we will make sure it is more clear.
>
> > Table 1: There is no result about OpenWebText dataset.
>
> Thank you for noticing this. Due to limited compute resources, we could not complete the LTH experiments on OpenWebText dataset. We were only able to complete the quaternion training. We have now removed it from the paper to avoid confusion.
>
> > (typo) line 259: “CIFAR” -> “MedMNIST”
> > (typo) line 227: Prune rate might be “0.3, 0.5, and 0.3” -> “0.3, 0.5, and 0.7”
> > The reference page needs to be rearranged
>
> Thank you for pointing those out. We have corrected the typing errors. However, we do not understand “The reference page needs to be rearranged”. Please provide us more details so that we can fix the same.
>
> Please refer to the revised manuscript to see the updates.

---

### Official Review · Reviewer_iDxz · 2024-10-08
**Review of Transformers at a fraction**

**Confidence:** 2

**Summary:**

The authors investigate the accuracy of quaternion transformers and the accuracy of Lottery Ticket Hypothesis (LTH) in pruning (quaternion) transformers. Specifically the Vision Transformer (ViT) and nanoGPT architectures are investigated.

**Strengths:**

The authors performed an extensive evaluation of their proposed approach on existing transformer architectures.
Through experimental validation, the authors show that incorporating quaternions in a transformer architecture and performing pruning using LTH increases the computational efficiency while maintaining performance.
Furthermore, the implementation is available open source as a PyTorch package.

**Weaknesses:**

It would be better to formulate the 4 research questions as research contributions.
Then it is also easier to emphasise the novelty in this paper. For example, quaternion transformers exist. Does this paper mean that this extensive evaluation is not available in literature? If this evaluation is done here for the first time, it could be more emphasised.
Also in the literature study in this paper, no references are made to existing Quaternion Transformers networks.

Idem for the Lottery Ticket Hypothesis there are gaps in the literature study, for example: https://arxiv.org/abs/2005.03454
Does this paper tackle an additional problem related specifically to the ViT or nanoGPT architecture?

**Justification:**

I am not in expert in the current state of the art regarding Transformers, but I think the paper overall contains an interesting contribution that is worth sharing. Although I would say the current shortcomings needs to be addressed before the paper can be accepted.

---

> ### Author Rebuttal · Authors · 2024-10-22
>
> We thank the reviewer for their valuable feedback.
>
> > It would be better to formulate the 4 research questions as research contributions. Then it is also easier to emphasize the novelty in this paper. For example, quaternion transformers exist. Does this paper mean that this extensive evaluation is not available in literature? If this evaluation is done here for the first time, it could be more emphasized.
>
> Thank you, we agree with the reviewer's suggestion. We have now formulated the research contributions (earlier research questions) accordingly as advised.
>
> > Also in the literature study in this paper, no references are made to existing Quaternion Transformers networks.
> > Idem for the Lottery Ticket Hypothesis there are gaps in the literature study, for example: https://arxiv.org/abs/2005.03454 Does this paper tackle an additional problem related specifically to the ViT or nanoGPT architecture?
>
> Thank you for the suggestion. We have added the following references for the following papers in the Introduction Section.
>
> **Quaternion Transformers:**
>
> 1. Zhou, Y., Guo, L., & Jin, L. (2023, June). Quaternion Orthogonal Transformer for Facial Expression Recognition in the Wild. In ICASSP 2023-2023 IEEE International Conference on Acoustics, Speech and Signal Processing (ICASSP) (pp. 1-5). IEEE.
> 2. Tay, Y., Zhang, A., Luu, A. T., Rao, J., Zhang, S., Wang, S., ... & Hui, S. C. (2019). Lightweight and Efficient Neural Natural Language Processing with Quaternion Networks. In Proceedings of the 57th Annual Meeting of the Association for Computational Linguistics. Association for Computational Linguistics
> 3. X. Yang, W. Cao, Y. Lu and Y. Zhou, "QTN: Quaternion Transformer Network for Hyperspectral Image Classification," in IEEE Transactions on Circuits and Systems for Video Technology, vol. 33, no. 12, pp. 7370-7384, Dec. 2023, doi: 10.1109/TCSVT.2023.3283289.
>
> **Lottery Ticket Hypothesis on Transformers:**
>
> 1. Chen, T., Frankle, J., Chang, S., Liu, S., Zhang, Y., Wang, Z., & Carbin, M. (2020). The lottery ticket hypothesis for pre-trained bert networks. Advances in neural information processing systems, 33, 15834-15846.
> 2. Brix, C., Bahar, P., & Ney, H. Successfully Applying the Stabilized Lottery Ticket Hypothesis to the Transformer Architecture.
> 3. Shen, X., Kong, Z., Qin, M., Dong, P., Yuan, G., Meng, X., ... & Wang, Y. (2023, August). Data level lottery ticket hypothesis for vision transformers. In Proceedings of the Thirty-Second International Joint Conference on Artificial Intelligence (pp. 1378-1386).
>
> [Brix et al.](https://arxiv.org/abs/2005.03454) increases stability by using Magnitude Pruning (MP) over the Stabilized Lottery Ticket (SLT) and experiments are performed only on the translation task.
>
> In our paper, we try to find lottery tickets in quaternion transformers and showcase that quaternion lottery tickets are more stable in extreme sparsity as compared to the real counterparts.
>
>
> Please refer to the revised manuscript to see the updates.

---

### Meta-Review · Area_Chair_77ov · 2024-10-30

**Recommendation:** Accept (Oral)
**Confidence:** 5

**Metareview:**

In this paper the authors show that quaternion neural networks and sparse networks created by pruning weights can reduce the number of parameters in transformer-based models, making them easier to run on devices with smaller memory. Experiments on vision and language tasks show that pruned quaternion models perform better than real-valued models in severely sparse conditions. The proposed method is implemented in a PyTorch package called qytorch.

Quaternion variations in Transformer models reduce weight parameters by one-fourth compared to real-valued Transformers. Pruning these Quaternion-based models using the Lottery Ticket Hypothesis maintains performance comparable to real-valued Transformers. Combining them with other techniques like quantisation and pruning can lead to more efficient and deployable models.

These are some neat results worth publishing and presenting at the conference.

**Suggested Changes To The Recommendation:**

1: I agree that the recommendation could be moved down

---

### Decision · Program_Chairs · 2024-11-06

**Decision:**

Accept (Oral)

**Comment:**

We recommend an oral and a poster presentation given the AC and reviewers recommendations.